# On-Line Laser Triangulation Scanner for Wood Logs Surface Geometry Measurement

**DOI:** 10.3390/s19051074

**Published:** 2019-03-02

**Authors:** Piotr Siekański, Krzysztof Magda, Krzysztof Malowany, Jan Rutkiewicz, Adam Styk, Jakub Krzesłowski, Tomasz Kowaluk, Andrzej Zagórski

**Affiliations:** 1Institute of Micromechanics and Photonics, Warsaw University of Technology, 8 Św. A. Boboli St., 02-525 Warsaw, Poland; k.magda@ksmvision.pl (K.M.); j.rutkiewicz@mchtr.pw.edu.pl (J.R.); a.styk@mchtr.pw.edu.pl (A.S.); j.krzeslowski@mchtr.pw.edu.pl (J.K.); 2KSM Vision Sp. z o.o., 9/117 Sokołowska St., 01-142 Warsaw, Poland; k.malowany@ksmvision.pl; 3Institute of Metrology and Biomedical Engineering, Warsaw University of Technology, 8 Św. A. Boboli St., 02-525 Warsaw, Poland; t.kowaluk@mchtr.pw.edu.pl; 4Faculty of Materials Science and Engineering, Warsaw University of Technology, 141 Wołoska St., 02-507 Warszawa, Poland; andrzej.zagorski@pw.edu.pl

**Keywords:** wood logs geometry measurement, laser triangulation scanner, calibration

## Abstract

The paper presents the automated on-line system for wood logs 3D geometry scanning. The system consists of 6 laser triangulation scanners and is able to scan full wood logs which can have the diameter ranging from 250 mm to 500 mm and the length up to 4000 mm. The system was developed as a part of the BIOSTRATEG project aiming to optimize the cutting of logs in the process of wood planks manufacturing by intelligent positioning in sawmill operation. This paper illustrates the detailed description of scanner construction, full measurement process, system calibration and data processing schemes. The full 3D surface geometry of products and their applied portion of selected wood logs formed after cutting out the cant is also demonstrated.

## 1. Introduction

A very important element in industrial practice, in the entire technological chain, is control and constant monitoring of the quality of raw materials for producing final products. This is closely related to the application of more and more modern measuring techniques strictly adapted to specific demands of the measured objects. In this context, particular attention should be paid to the wood industry, where natural resources, renewable in a long time (counted in decades) should be used in the most economical way possible. One of the main problems in sawmill tracks, particularly in the initial phase of material processing, when cutting oval wood logs onto rectangular boards, is the reduction of generated waste, directly translating into maximization of the obtained material. In current practice, it was a sawmill track operator who was setting wood log to the first cut on the basis of his knowledge and experience. However, such an approach is not optimal due to the rather subjective assessment of the best position of the log on the sawmill track. That is why there is a constant search for the methods of the objective assessment of the proper log position on the sawmill track. For this purpose, it is inevitable to know the shape of the surface of the entire wood log. Having a 3D log model, due to the use of optimization algorithms, it is possible to generate the best cutting planes of the measured log, and, as, a consequence—obtain the maximum capacity of the usable material for further processing in the cutting process. 

General information on geometrical parameters of a given log (its length, upper, lower and middle diameter and capacity) can be obtained from manual measurements according to the guidelines published by authorized institutions and state administration offices or accepted national standards. Manual measurements are taken with the use of calibrated linear gauges. However, obtaining complete quantitative information on the shape of the log surface is only possible by using different measuring techniques imaging the shape of the surface. For such purpose, non-contact optical measuring techniques are applicable. For wood logs measurement, laser scanners are used for the most part, providing satisfactory accuracy, as well as scanning speed. The simplest solution applied in the wood industry are systems containing a single measuring system and allowing for measuring only the transverse diameter of the log projection (usually vertical) [1,2]. The development of such systems are the systems using 2 orthogonal scanners and measuring 2 orthogonal diameters in one cross-section of the log. In order to regain information on the entire dimension of the log, at least 4 laser scanners are required. Such systems are commercially available (see. e.g., JoeScan JS-25 X-SERIES [3] or HERMARY DPS-4024VE or DPS-4024HA [4]) and may be incorporated directly into the production chain. For example, Porter Engineering Ltd. of Richmond, S.C., Canada has developed real time software, Real Time for Real Trees with real Shape (RT3) for shifting, skewing and tilting canters, end-doggers, sharp chains, Chip-N-SawsTM and twin and quad band primary breakdown systems [5]. These programs feature full graphic displays for the operator and accurate true shape computer modeling of each log in three dimensions. A special version of RT3 is used to automatically rotate the log for an additional increase in recovery. RT3 has also been developed for optimum bucking applications, taking into account Sweep, Crook, and Rotation. The most sophisticated scanning systems can now scan logs for shape and provide full three-dimensional images from which computers can calculate the best rotational axis and optimum sawing solution for each log [6]. There have been significant developments in scanning systems for the optimization of board edging and trimming. Scanning is now being done at production speed, and the trend is to maximize profile resolution and measurement accuracy, while minimizing the number of scanning components per profile. Gains in yield are substantial. For example, an edger optimizer can consistently achieve 85-95 percent volume recovery at speeds up to 200 m/min [7]. 

In the wood industry, more advanced measuring systems are also used allowing for the measurements of the internal structure of the logs, including defects and delamination of the internal material, as well as allowing for the measurement of mechanical properties of the material. The most widespread internal imaging techniques of the wooden material include ultrasound, CT scan or MRI scan [8]. Often, in the case of material tests, microwave and thermal imaging techniques are also used [8]. With the advance of scanning equipment, it is possible to develop automated systems that aid bucking and sawing processes. Researchers have experimented with technologies and methods that locate and classify internal, as well as external defects on either softwood or hardwood. Systems including those using X-ray/Computerized Tomography (CT), X-ray tomosynthesis, magnetic resonance imaging (MRI), microwave scanning, ultrasound, and ground-penetrating radar have been researched and developed to detect internal defects [9,10,11,12]. Advanced computer vision and image-processing methods and algorithms were proposed for internal defect inspection on hardwood [13,14,15]. Some of these methods and technologies provide high detection rates as their data have extremely high resolution. However, issues, such as high cost, low processing speed, data instability, and environmental safety keep them from being commercially available. Recently, Microtec introduced the first real time CT log scanner [16]. High-resolution 3D laser-scanned data of hardwood logs show minute surface variations, making it feasible to detect external defects. Compared to other systems, such as X-ray/CT, laser-scanning equipment is more economical, safer, faster, and more suitable for log sizes [17]. 

The article presents the results of research carried out for the needs of Barlinek—the largest Polish and European manufacturer of multi-layer floor boards. The works concerned the increase of the efficiency of the wooden material in the production process of floor boards. As a result of works described in this article, a system for measuring the shape of the wood logs, made up of a scanner equipped in 6 measuring heads (based on the laser triangulation method) and dedicated software for data analysis allowing for the optimization of the log cut was provided. 

## 2. Review of Optical Methods for Shape Measurements

As mentioned above, during the implementation of the described works, the authors decided to develop a measuring device, whose main aim is to map the entire shape of a 3D wood log on a sawmill track and create its numerical model, and then—the most optimal positioning and cut of this log on the sawmill track.

In general, for the needs of measurement of the 3D objects surfaces, it is possible to use different measuring techniques that differ in the time of data acquisition, measurement accuracy or complexity of the measuring system. Knowing the requirements regarding metrological parameters of the measuring device, imposed by the technological process realized at Barlinek, an overview of the industrially available, non-contact methods of 3D objects imaging for the purposes of wood logs scanning was performed.

3D scanning can be performed with the use of numerous technologies. Their general classification assigns them to 2 categories: contact and non-contact. In contact methods, a direct contact with the measured object needs to be provided, that is why these methods lose popularity in favor of non-contact methods, which additionally, due to the applied type of light can be divided into active and passive methods. Passive methods make use of unmodulated (static) lighting, which is registered by individual cameras or their complex systems. Due to this approach, passive methods are inexpensive and easy to calibrate. In the systems using active methods, specially selected light sources emit radiation, which is modulated (temporally or spatially), and then received and analyzed by detectors. Thanks to this approach, active methods are more accurate.

### 2.1. A method with Structural Lighting

3D scanning utilizing a structural light projection technology is a relatively new measuring method. Its rapid development occurred with the appearance of digital technology of image projection and acquisition. At the moment, numerous variations of the described method have been proposed allowing for, inter alia, real-time work with a different number of the registered images, as well as different accuracy depending on the examined object [18]. In general, the method with structural lighting consists in projecting raster images on the surface of the examined object by the projector or laser with a proper optical system. These images, when meeting the examined object, become deformed and such changes are observed by at least one camera in the measuring system. Structural lighting enables the use of any raster pattern provided that the analysis of its deformation will allow to arbitrarily define the geometry of the surface of the measured object.

The most common method used for the analysis of the distorted fringe images is a method with temporal phase shift, which requires registration of at least 3 fringe images (the more of the registered images, the more accurate the measurement), where spatial distribution of the fringes is changed/modulated in a controlled way [19,20,21]. A complementary method with a binary raster projection in the form of a Gray code requires projecting and registering a sequence of well-defined binary patterns (the two subsequent patterns differ only with the state of the bit) [22,23]. When measuring dynamic objects or when it is impossible to sustain stable experimental conditions during the registration of more than one fringe image, the method utilizes surface geometry calculations from a single image with a projected raster. The method of Gray codes projection is not used at that time. For the analysis of individual images with sinusoidal raster, the so-called one-image algorithms are used, such as Fast Fourier Transform (FFT) [24,25,26], spatial carrier phase shifting (SCPS) [27] or Wavelet Transform (continuous -CWT and discrete—DWT) [28,29,30]. 

The general advantage of raster projection is the high spatial resolution of the measurements; however, the periodic character of the projected patterns can lead to indetermination in decoding the information on the shape of the object (it depends, among others, on the used computational algorithms). An additional asset of the described method is the simple structure of the system, not requiring movable mechanisms and in the case of using standard projection and imaging systems—low price of the components. Application of a digital projector (incoherent light) also eliminates the influence of the spot noise effect. The drawback of this solution is the difficulty in registering the reflecting, non-uniform and discontinuous objects. An exemplary measuring system utilizing structural lighting technique used for measuring the shape of a wood log is presented in Figure 1.

### 2.2. Laser Scanning Method (Laser Triangulation)

Laser triangulation is classified as a non-contact active measuring method, where the source of radiation is a laser and a receiver—a CCD/CMOS camera with a properly selected lens (Figure 2). Laser triangulation method in its measuring concept is extremely simple. The triangulation system is composed of a light source, a photosensitive detector and a measured object. They are placed on the plan of a triangle constituting its vertices [31]. The light source, i.e., the laser, illuminates the surface point-wise or linearly and the projections are observed by the detector (a CCD camera most likely) which is placed at a set angle towards the laser axis. Due to providing this perspective, it is possible to determine the position of the measured points, and as a consequence—the dimensions of the object. In the case of a laser generating a line illuminating the examined object (the system most commonly used in machine vision) the shape of the object is encoded in the deformations of this line on its surface. Despite the simplicity of the operating principle, the measurement with the use of laser triangulation requires the implementation of the detailed software connecting the parameters of the components of the system. The implementation of the code is determined by the way of the setup of a laser triangulation selected to optimize the measurement. Optimization depends on the measured object, its type, dimensions, position and the required accuracy of scanning in all directions, scanning speed and the type of the used scanner.

Measuring systems based on triangulation are offered mainly on two basic setups—portable, with a measuring arm and a mounted one, integrated with the production line. Depending on the type of laser scanning system, as well as its application the accuracy of measurement may vary. The most accurate systems can reach up to few µm in accuracy, whereas typically laser triangulation systems accuracy is in the level of 20—200 µm [32].

### 2.3. ToF—Time of Flight Systems

The systems (cameras) applying ToF are the devices allowing for 3D imaging on the basis of the known speed of light and the determined time of beam propagation between each point of the image and the matrix detector [33]. ToF systems are classified as active methods, where a modulated light signal is generated from the light source, which is registered and analyzed after illuminating the object. ToF systems do not require a spatial scanning procedure, so they are classified as optical field methods, where the information from the entire measuring field is registered in a single image. The sequence of the collected images carries information on the dynamics of the object. The operating scheme of ToF systems is presented in Figure 3.

ToF systems became widespread in public use around 2000, when electronic devices which are their basic operating principle have reached the required processing and signal transfer speeds. ToF systems are characterized by a measuring range from single centimeters to many kilometers. The standard resolution of the measurement is about 1 cm, while the transverse resolution of the measurement depends on the resolution of the used matrix detectors used in measurements. Usually, the applied matrix detectors have a resolution much below 1 Mpix. In comparison to laser scanners they are faster in collecting data. In many solutions, the speed of collecting images is over 1000 fps [34]. 

ToF systems utilize different solutions concerning light signal modulation and processing information registered by the detectors. Among them, the following groups may be distinguished: RF modulation and phase detectors illuminators, systems with time gating and systems with the direct time of flight measurement [35]. The first group includes a high resolution modulated harmonic system (lighting with variable sinusoidal intensity at a specified speed) and the phase shift between a signal emitted by the source of light and received by the detector are analyzed/determined. In these solutions, the determined phase distribution carrying the information on the shape of the examined object is obtained in the form of a modulo 2π what requires an additional unwrapping procedure in order to regain the correct information. The second group of solutions makes use of the embedded gating system (electronic shutter) operating at the same frequency as the illumination system generating light impulses with a particular frequency. Since in such a system, impulse generation frequency and their gating is equal, the amount of light registered by the gating detector is proportional to the path travelled by the impulse and is the basis for calculating the distance between the object and the camera, and as a consequence—the shape of the measured object. ToF systems with time gating allow for obtaining even sub-millimeter measuring resolution. Systems from the third group measure the direct time between the generation of the illuminating impulse and its registration in each pixel on the matrix detector. This procedure is also known as the trigger mode allowing to get the total time and space information with the use of a single illuminating impulse. Such an approach enables instant registration and quick processing of information on the measuring scene.

ToF systems have significant assets when compared to other 3D imaging techniques. These are: simple construction—the systems are very compact in relation to triangulation or stereovision systems. The illuminating systems and the detectors are placed next to each other and, unlike other systems, do not require any base distance between them. What is more, these systems do not require any scanning techniques nor devices to register a 3D image. They utilize efficient algorithms of data processing. The process of determining information on the distance between the object and the detector is fast, due to the simple computational algorithm, and usually performed in the detector, which requires very little computing power. An unquestionable asset is also the measuring speed. ToF systems make a measurement in a measured scene with the use of a single image. Taking into account the operating speed of the detectors used in these systems, it is possible to use the same systems in real-time applications.

Among the disadvantages of ToF systems is the limited spatial resolution of the measurement (in comparison to other 3D imaging techniques), related to the limited resolution of matrix detectors used in these systems, as well as the limited resolution of the shape measurement related to the accuracy of generation and processing of electronic control signals. In these systems, the impact of the background illuminating the measured object is important. In numerous cases, the radiation coming from outside the illuminating system in ToF is also registered by matrix detectors, which significantly impedes the processing of the signal lowering the SNR ratio. This is the case, e.g., in sunlight, which often after passing through a banded color filter can still have a power density higher than the density of the radiation power from the illuminating system. This effect may be minimized in the systems where the techniques of quick electronic gating are used. Another problem is interference in the case of multi-camera ToF systems. The systems illuminating parts of the object placed next to each other may cause mutual leaks of the signal to the adjacent detectors, what significantly affects the results of the measurement. In such cases, special systems of time multiplexing or different frequencies of the light modulation are used. In such systems, multiple reflections can also be problematic. Unlike laser scanning systems, in ToF systems the entire measuring scene is illuminated. For phase measurement systems due to multiple reflections, a single pixel can be reached through different paths, that is why the determined distance can be slightly larger than the real one. In the direct time of flight systems, the measurement of the mirror surfaces can cause a similar effect.

### 2.4. Structure from Motion (SfM)

It is a type of passive optical measuring method consisting of the reconstruction of a 3D object on the basis of a set of its images taken at different distances and at different viewing angles [36]. The higher the number of the images, the more accurate the 3D shape is obtained. However, it significantly extends the time of the image analysis. There is also an active version of this method, where additional sources of light project markers on the examined object. The operating principle of the SfM method is schematically presented in Figure 4.

After completing the images, special algorithms are used (usually SIFT—Scale Invariant Feature Transform [37,38]) identifying specific points of the object occurring in the subsequent images, which are then attributed by the descriptors. Then, using the bundle adjustment algorithm [39,40], the position of the virtual camera towards the object is estimated and a sparse 3D point cloud is created. To obtain a dense 3D point cloud, additional CMVS (Clustering View from Multi-view Stereo) algorithm [41,42] is used, which put the images under clustering, followed by PMVS2 (Path-based Multi-view Stereo) algorithm [43] which reconstructs 3D information collected from the clusters.

The advantage of this method is a simple and inexpensive measuring system and a wide measuring range. Unfortunately, it only works for objects with many characteristic points related to color or shape. Otherwise, it can cause a lot of errors. Therefore, this technique is often used only for the purposes of visualization. To obtain reliable results, it is necessary to know the parameters of the used camera and the consistency of ambient conditions (mainly lighting), and then—accurate calibration allowing for associating the 3D model dimensions with the corresponding real dimensions of the object (scaling).

The objects measured with the SfM method should have a dispersive surface with a high number of characteristic points resulting from their shape and color. The objects with such topography will be reconstructed more accurately than those with a uniform surface. This is one of the drawbacks of this method. On a single 3D model, the areas with different degrees of the mapping quality may be obtained. In relation to laser triangulation methods, time of flight and fringe projection, the mapping itself is exposed to high errors resulting from the automatic operation of the algorithms on unscaled data. For that reason, at the final stage of processing, user verification is necessary. The advantages of the SfM method are the ability to measure objects in a huge range of dimensions—from a few centimeters to buildings and even cities [44]. 

### 2.5. Comparison of the Methods

Table 1 contains a summary of the most important parameters of all previously described methods. All mentioned methods are used to measure the topography of the objects in 3D coordinates. The quality of the topography of the surface mapping mainly depends on the resolution and accuracy of the measurement. From the aforementioned methods, the method utilizing structural lighting in the form of sinusoidal fringes has the best resolution. It is able to achieve subpixel resolution, reaching even 1/20 pixel size. In the metric representation it is 5 µm. The other methods are one class worse in that respect. The accuracy of measurement is dependent on the size of the measuring field. For active scanning methods described in Table 1, the measuring field is similar and usually ranges from tens of centimeters to 2 m. The measurement of larger objects is ineffective due to the fact that radiation sources have limited power. In the given ranges of measuring fields, both methods can achieve the accuracy of 1/1000 of the size of this field, but in the case of laser triangulation, the maximum accuracy will be about 50 µm and will not be exceeded with the drop in the dimensions of the measuring field due to the finite width of the laser line. Time of Flight method is a technique with the lowest resolution from among all the analyzed methods, what significantly affects the accuracy of the measurement in a negative way. Its greatest asset is the high speed of the measurement. On the contrary, with the SfM, which is a passive method, there are no obstacles that would impede the measurement of the objects on a large size scale. In this method, determining the accuracy is extremely difficult, because it depends on the size of the measuring field and the surface of the measured object. The more characteristic points on the measured surface, the higher accuracy can be achieved. That is why the SfM method seems to be less beneficial when juxtaposed with the three other methods, in which the sampling of the geometry is constant and independent of the element’s examined surface. This allows to predict the results of the measurement and achieve high repeatability.

In all methods, acknowledging the properties of the measured object, it is only possible to measure the objects with the dispersive and non-transparent surface, and in some cases—partially non-transparent. The color reproduction is possible in any of the aforementioned techniques, but in the case of laser triangulation, structural lighting and Time of Flight methods, a color detector is required. To reproduce the true color of the surface, it is required to illuminate the scene constantly. It also affects the obtained data. This parameter is particularly important in structural lighting and SfM, where invariable lighting conditions are necessary to do get high measuring accuracy. In the case of laser triangulation and Time of Flight method, a well-chosen laser power makes the system independent of this factor. In terms of hardware requirements, SfM is the simplest technique. It is a representative of passive measuring methods thus not needing additional light sources, which takes place in the case of active methods included in the table (laser triangulation, Time of Flight—laser, structural lighting—projector).

All of the presented data are averaged values and can differ for individual uses. Depending on the particular aim, it is possible to choose a suitable measuring method. When scanning the entire perimeter of wood logs on a sawmill track, the solution utilizing laser triangulation systems seems to be the best. The speed of measurement and ensuring its proper accuracy, as well as a simplier construction of a system, with a possibility to easily separate the influence of the environmental disturbances (vibrations, dust, difference in temperatures, etc.) speak for its use. 

## 3. Configuration of Laser Triangulation Scanner

The shape scanner enabling shape measurement of the wood log consists of six components of laser triangulation, further referred to as measuring heads (Figure 5). Each head is composed of 4 elements (Figure 6): A laser ruler (Z-LASER ZM18), a camera (Automation Technology C2 Series Model C2), 2 optical mirrors and a supporting structure made of aluminum profiles. The optical axis of the camera and the laser is set at an angle of 30 degrees. The use of optical mirrors allowed for reducing the dimensions of the device and facilitate the alignment of the laser beam position and the camera’s field of view. Each component of laser triangulation is connected to the computer equipped with an Intel Core i7 3610QE 2.30 GHz processor and 8 GB RAM. The system can be seen in Figure 7.

## 4. Measurement Process 

During the measurement, data from each camera is processed in a separate thread. Such approach provides possibly small delays in data acquisition and processing. Processing data from a single system goes as follows:
Acquisition of the profileExtraction of the profileFiltering of the profileCalculating the point cloud from the profile in the local coordinate system of the cameraTransformation of the point cloud into the global systemIncrement of the displacement of the global coordinate system in X-axis


### 4.1. Acquisition of the Profile

On the signal from the microcontroller, each of the cameras starts to acquire the frame in accordance with the exposure time set previously. Synchronization through an external system triggers the cameras at the exactly same time regardless of the current task of the host computer.

### 4.2. Extraction of the Profile

In each column of the image, a point corresponding to the position of a laser ruler on a camera matrix is searched. Depending on the user’s choice, it can be determined either as the point with the highest intensity in a given column of the image or as the center of gravity of the pixels of a laser ruler, where the intensity in a given pixel is the weight. The result of this process is a set of x, y pixels on the matrix.

### 4.3. Filtering of the Profile

The calculated profile needs to be filtered due to the fact that noises may appear in the system caused by the reflections of lasers from the body of a scanner and pane, and those caused by the influence of external lighting. A two-stage procedure of data filtering was proposed eliminating the noise and filtering the elements of a scanner and a conveyor. It was decided to filter profiles in the field of image due to the fact that they can be significantly faster than data filtering in 3D space.

In order to filter the noise, a one-dimensional filtering of the pixels is performed in a window with the width of *n* selected by the operator. For each pixel of the profile, the number of pixels in a window is verified and when it is lower than the threshold *m*, a given pixel is considered noise, and eliminated from the profile.

Next, each pixel of the profile after filtering is compared to the value of the mask in a given column. If the distance in the *y*-axis between the analyzed pixel and the value of the mask in a given column is smaller than the threshold adopted by the user, a given pixel is considered a measurement of the scanner or conveyor casing and rejected. The mask is made on the conveyor without any load, and due to the large weight of the scanned logs, the conveyor’s profiles can bend and hence the need for a non-zero threshold. 

### 4.4. Calculating the Point Cloud from the Profile in the Local Coordinate System of the Camera

For each pixel of the profile, a ray is created coming through a nodal point of the camera and a given pixel on the matrix and its intersection with a corresponding laser plane is studied in a local coordinate system of the camera. Intersection is a point belonging to the log. The result of this process is a collection of 3D points corresponding to the observed profile at a given moment.

### 4.5. Transformation of the Point Cloud into the Global System

In order to complete the profile of a log at a given moment, each set of the points has to be transformed from the local into the global coordinate system calculated previously during the global calibration procedure. The result of this process is a complete profile of a log at a given moment.

### 4.6. Increment of the Displacement of the Global Coordinate System in X-Axis

In order to get a full geometry of a log, the current value of a X coordinate of the log axis needs to be shifted in a global coordinate system by the value read from the laser rangefinder or by the product of the time set between subsequent triggers of the cameras and the known speed of log relocation. 

## 5. Scanner Calibration

The scanner consists of 6 laser−camera pairs that form 6 single laser triangulation systems.

Calibration of the system can be divided into 2 stages:
local calibration of each laser triangulation systemglobal calibration


### 5.1. Local Calibration

Local calibration of each system requires determining the calibration of the camera and a relative position of the laser ruler. In order to calibrate the camera, several frames of the calibration pattern should be collected at different distances and angular settings with respect to the camera. In each frame, centers of the circles are determined, making the basis for calibration of the camera (Figure 8).

#### 5.1.1. Calibration of the Camera

Calibration of the camera is a process of determining the parameters of the lens and imaging plane. The specified parameters may be used to correct the distortion of the lens or to determine the position of the camera in relation to the object with a known geometry on the basis of a single image. Calibration of the cameras is commonly used, e.g., in machine vision, robotics, 3D scanning and reconstruction.

Within the studies, the algorithm of camera calibration with the use of a flat pattern proposed by Zhang [45] was applied. The final effect of the calibration process is a matrix of internal parameters of the camera ($f_{x},\;f_{y},\;c_{x},\;c_{y}$) and 5 distortion coefficients ($k_{1},\;k_{2},\;p_{1},\;p_{2},\;k_{3}$).

#### 5.1.2. Distortion Coefficients

Hence the pinhole model of the camera does not have a lens, it also does not acknowledge distortion introduced by the optics of the real camera. For this reason, it is necessary to extend the calibration procedure and determine the coefficients of the lens distortion. The calibration algorithm takes into account the radial distortion caused by the curvature of the camera lens and the tangent related to the inaccurate assembly of the camera sensor plane. 

Radial distortion
(1)$$x^{\prime }=x\left(1+k_{1}r^{2}+k_{2}r^{4}+k_{3}r^{6}\right)$$
(2)$$y^{\prime }=y\left(1+k_{1}r^{2}+k_{2}r^{4}+k_{3}r^{6}\right)$$
x,y—coordinates before radial distortion correction$x^{\prime },y^{\prime }$—coordinates after radial distortion correction $k_{n}$—*n*th radial distortion coefficientr— the radius of the circle with the center in the upper left corner of the image passing through the point (x,y) $r^{2}=x^{2}+y^{2}$


Tangential distortion
(3)$$x^{\prime }=x+\left[2p_{1}xy+p_{2}(r^{2}+2x^{2}\right)]$$
(4)$$y^{\prime }=y+\left[2p_{2}xy+p_{1}(r^{2}+2y^{2}\right)]$$
$p_{n}$
—*n*th tangential distortion coefficient


The matrix of the camera

After correcting the distortion, camera images are used to determine the matrices of the internal parameters of a pinhole model of the camera.
(5)$$\left[\begin{array}{c} x\\ y\\ w \end{array}\right]=\left[\begin{array}{ccc} f_{x} & 0 & c_{x}\\ 0 & f_{y} & c_{y}\\ 0 & 0 & 1 \end{array}\right]\left[\begin{array}{c} X\\ Y\\ Z \end{array}\right]$$
$f_{x}$, $f_{y}$—focal length in x and y axes of the camera in pixels $c_{x}$, $c_{y}$—coordinates of the centre of projection on the camera matrix 


The matrix of the camera maps the point (X,Y,Z) onto the image plane (x,y). Coefficient w is in this case equal to Z and is caused by the homogeneity between the flat calibration model and the flat camera image.

The determined coordinate system of the camera has its center at a nodal point, with *X* and *Y* axes in accordance with the image plane and the *Z* axis is directed from the camera.

#### 5.1.3. Calibration of the Laser with Respect to the Camera

In order to determine the position of the laser in the camera’s coordinate system, a few pairs of image, and the profile have to be collected. In order to collect a pair, an image should be collected previously, and then keeping the pattern intact switch on the laser and collect the profile (Figure 9).

Next, from the pixels belonging to the profile, a straight line is determined in the image using the RANSAC algorithm. Then, for each pixel belonging to the laser line, a point is determined in the camera’s coordinate system, which is the intersection of the ray from a given pixel and a pattern plane, whose position relative to the camera is specified thanks to the calibration of the camera. 

Next, with the use of the least squares method a plane is adjusted to the set of the points from several registered laser profiles. This way, the position of the laser plane is determined in the camera’s local coordinate system. 

### 5.2. Global Calibration

The aim of the global calibration is finding mutual transformations between particular cameras. For this purpose, a hexagonal artifact with a radius of 232 mm was designed and made. A different set of Charuco [46] codes was fixed to each of the six walls (Figure 10). The global coordinate system is put in the geometric center of the calibration artifact and its axes are directed, *x*—along the conveyor; *y*—right; and *z*—up. As the mutual positions of the walls of the artifact are known, it is possible to determine the position of each camera towards the pattern. We use ePnP algorithm [47] to determine the camera pose based on 2D-3D correspondences between chessboard corners and their 3D coordinates. Due to the fact that the center of each local coordinate system is in the nodal point of the camera we are able to determine the mutual transformation of local coordinate systems.

In this way, a preliminary adjustment of local systems was achieved, but it was necessary to optimize them in terms of noises related to the determination of the markers and the inaccuracy of determining the calibration of the camera. For each camera, a rotation matrix was calculated that maps the local *x*-axis in the world coordinate system on the *x*-axis of the global coordinate system. It results in a slight shift of the cameras and ensures angular compatibility of the particular directional scans (Figure 11).

#### Preparing the Masks

The structure of the conveyor with two steel profiles supporting the log passing through the scanner makes them cover a part of the log during scanning. Such profiles should be filtered from the further scans together with possible scanner components that can be scanned if they are not covered by the log. A procedure was proposed that includes making *n* scans with no log on the conveyor. Then, for each set of pixels p(n) belonging to the *n*th profile observed by each measurement head, the median of pixel values in a given image column is calculated and saved to the mask. After calculating the masks, the scanner is ready for operation. Example of the scanned log can be seen in Figure 12. 

## 6. Data Processing Procedure

The order of operations performed on the data generated by the scanner is depicted in the flowchart (Figure 13).

The cloud of 3D points obtained with the triangulation scanner is filtered globally and locally based on cylinders fitted to the whole cloud and consecutive parts of the cloud. Global filtering is aimed at eliminating noise points that are related to the scanning method (e.g., laser reflected off the scanner’s metal case), whereas local filtering is used in order to exclude unwanted log parts from further analysis as they could influence the results returned by the algorithm—the unwanted parts include mainly branch and bark leftovers.

After filtering, a plane is fitted to the cloud of 3D points as it has been observed that such a plane follows the direction of the net curvature of the log. It has been assumed that such a plane would be a good starting point for searching for the spatial orientation of the log that would yield best results in terms of volume of the cant fitted to the cloud.

The next step is to fit the products (i.e., planks) from the list supplied by the user along the spatial orientation derived in the way described above so that they form a cant. It is possible to either fit one type of product chosen by the user or try to fit all types of products and choose the best results in terms of volume automatically. The products are fitted by checking the collisions of cloud points with the cuboids representing respective product types. If there is no collision, i.e., no log point is contained within the cuboid, the plank is added to the cant. In case of collision, the cuboid is moved towards the center of the cylindrical coordinate system on a distance given by the user and the check is performed again. The cant fitting stops when no more planks can be added to the cant without collisions. For the purpose of the tests, it has been assumed that each plank’s length is approximately equal to the length of the scanned log and the cant consists of only one type of the product.

After the cant is fitted, the planks are also fitted to the parts derived after cutting the cant out of the log. For this purpose, a brute force approach to the knapsack problem has been applied. The algorithm stops either when the optimal solution has been found or when the specified amount of time has passed. 

After all the products have been fitted to the cloud of points, the angular and linear positions of the lasers marking the orientation of the cuts forming the cant are calculated. The cant edges are also drawn against the ends of the scanned log and saved to image files in order to help the operator to mark the log for cutting properly.

## 7. Experimental Results and Discussion

### 7.1. Local Calibration Assessment

For each triangulation system, a calibration described in Section 4 was performed. The accuracy of camera calibration is assessed based on the reprojection error defined as the mean value of the differences between extracted centers of the dots in the calibration pattern in image coordinates and projected its 3D metric coordinates onto the image plane. The accuracy of a laser-plane parameters computation is assessed based on RMS error between computed laser-plane and the 3D points extracted from each laser line. Table 2 shows the calibration results obtained for each triangulation system.

### 7.2. Scanner Validation

To estimate the uncertainty of the scanner, a custom-made ball-bar was made. This ball-bar consist of two bearing balls with a nominal radius equal to 25 mm connected with a 400 mm long steel bar. It was measured on ZEISS ACCURA measuring machine with maximum permissive error (MPE_E_) equal to 1,7 + L/333 µm, L in mm. Later it was scanned ten times in three different positions relative to the sawing plane: Parallel, perpendicular and oblique. In each scan the centers of the balls were determined and the distance between them was calculated. These tests were carried out according to ISO 5725-1:1994 standard and therefore they were examined using the same method, on the same item, in the same installation, by the same operator and using the same system within the short intervals of time. The results are summarized in Table 3.

Based on these measurements, we calculated the measurement uncertainty at the confidence level of 95% as 2 times maximum standard deviation, therefore the uncertainty of the scanner is equal to 2.16 mm.

In order to determine the accuracy of the scanner on example similar to a wood log, a measurement of a steel tube with a nominal diameter of 300 mm was performed using the examined scanner. The resulting point cloud was compared with the measurement taken using Metris MCA 2 articulated arm equipped with Metris MMCX80 laser scanner. The uncertainty of this setup is below 0.1 mm. Then, the mean and maximum matching error between the point clouds was determined. The results are included in Table 4. 

We achieved a maximum error of 8.02 mm which is higher than the maximum error obtained during ball-bar measurements. This may be caused by the measurement noise, reflections, etc. In general, high error values are most likely caused by inaccuracies in global calibration as can be seen in Figure 14. The points in yellow and red were measured by measurement heads 3 and 4 located at the bottom of the scanner. During global calibration a part of global calibration artifact is occluded by the conveyor (Figure 15) and that may cause the error in camera pose estimation and thus in transformation matrix from a local to a global coordinate system. After removing these two clouds the mean error dropped to 1.86 mm with a standard deviation equal to 1.49 mm. Bundle adjustment on camera positions and laser-plane parameters may be used to optimize global calibration and thus reduce the errors. We plan to implement it in the next version of the scanner. 

The cameras used in this scanner are designed to extract laser profile, not for photography, so the image that they deliver is affected by the noise (Figure 16). Due to the fact that the presented scanner must be able to work in the dusty environments, each triangulation system is protected by a pane. This pane may cause undesired reflections or distortions in the camera image. All mentioned factors may affect the accuracy of camera calibration and laser plane parameters. Nevertheless, the achieved overall scanner accuracy is sufficient for this kind of tasks. 

### 7.3. Sample Results

Sample log scanning results have been described below. It has been possible to obtain the cloud of points representing the log surface within the 320° angular range—the other part has been occluded by the bars on which the log had been transported. The tests performed indicate that the log scanning system works as expected.

The cloud of points generated by the scanner has been passed to the algorithm described in the previous paragraph, and thus the cutting scheme has been derived for a cant consisting of planks that are 170 mm wide and 31 mm thick (kerf of 4 mm has been assumed for the purpose of cutting simulation). The products fitted to the log are listed in Table 5 and visualized in Figure 17.

In the end, the processing time of a single profile was measured. All operations, except the determination of the log cut, take less than 10 ms in total, which leads to a conclusion that the presented scanner can work at 100 Hz. The time needed to calculate the log cut varies between 10 and 35 seconds depending on the number of the points of the log and the degree of the geometry complexity.

## 8. Conclusions

Automation of wood logs cutting procedure benefits in the efficiency of the wooden material in the production process of floor boards. It requires the precise knowledge of the shape of the wood logs processed in the mill. For this reason, the automated on-line system for wood logs 3D geometry scanning was developed. The system is based on 6 laser triangulation scanners (two sets of three scanners working in different wavelength regime) and is able to scan full wood log diameter ranging from 250 mm to 500 mm and up to 4000 mm long. The total scanning time for a single profile takes less than 10 ms making the system adaptable for sawmill track. Conducted calibration and validation of the system have shown its usefulness achieving the desired accuracy of 2.4 mm. Even better results can be achieved using different conveyor that allows full 360° measurement without occlusions. The developed system was tested at the largest Polish floor boards manufacturer sawmill showing its application potential in sawmill track. 

## Figures and Tables

**Figure 1 sensors-19-01074-f001:**
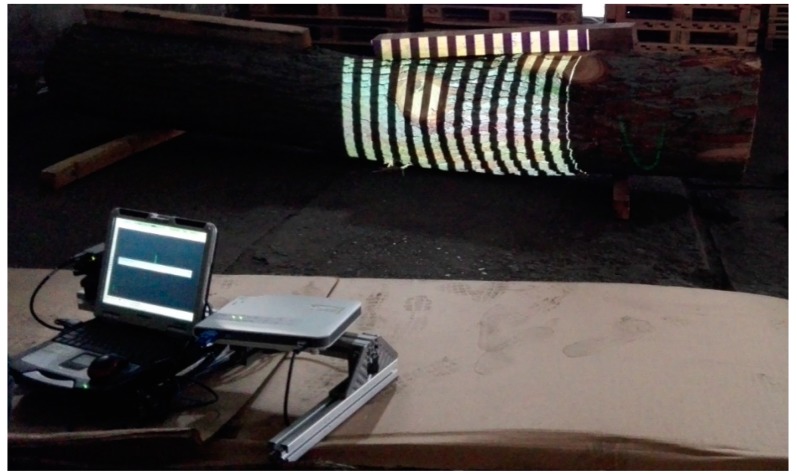
A measuring system utilizing structural lighting for examining the shape of the wood log surface.

**Figure 2 sensors-19-01074-f002:**
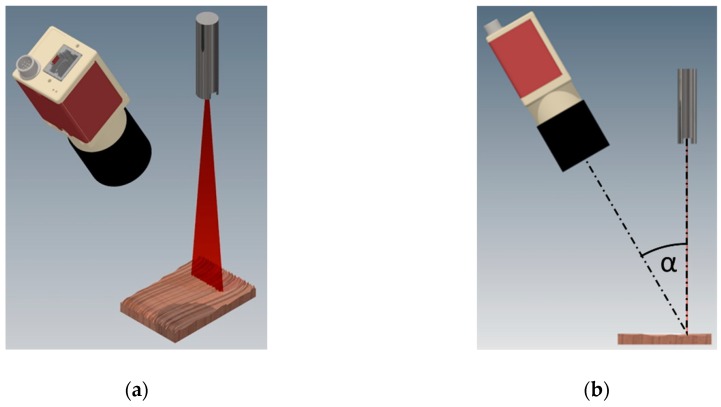
Visualization of the laser triangulation system, with laser line projected on the object and camera capturing pictures: (**a**) isometric view; (**b**) geometry of 3D vision system.

**Figure 3 sensors-19-01074-f003:**
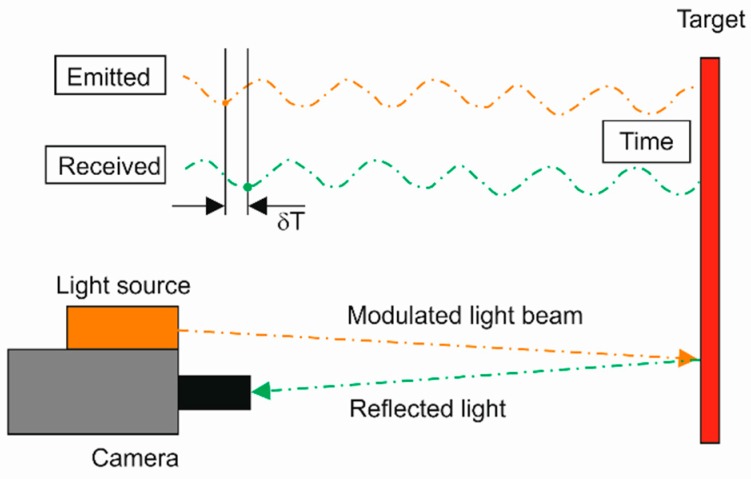
Operating scheme of time of flight (ToF) systems.

**Figure 4 sensors-19-01074-f004:**
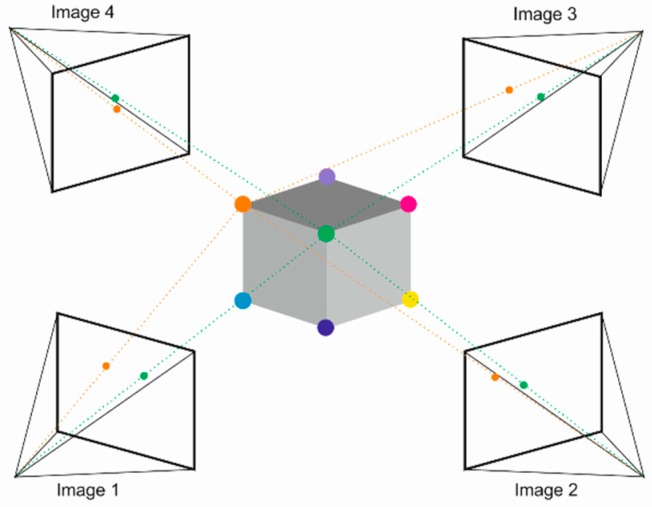
Operating principle of the SfM method—multiple images of the object captured from different angles.

**Figure 5 sensors-19-01074-f005:**
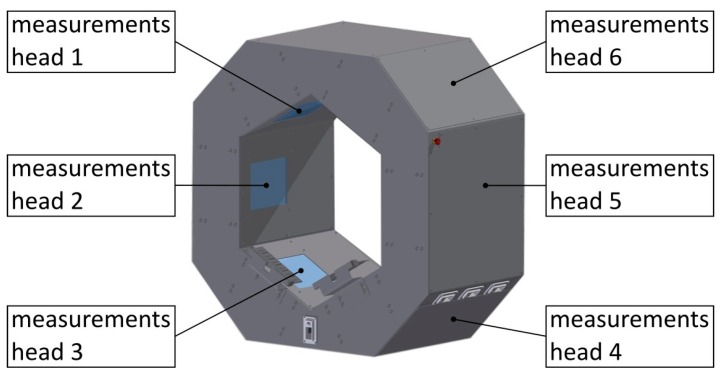
3D model of the laser triangulation scanner with measuring heads marked.

**Figure 6 sensors-19-01074-f006:**
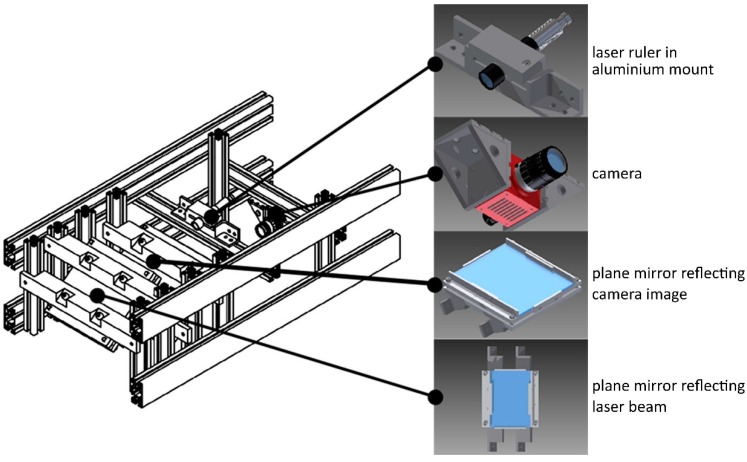
A construction of measurement head, the major components (laser, camera, mirrors) are mounted on a frame made from aluminum profiles.

**Figure 7 sensors-19-01074-f007:**
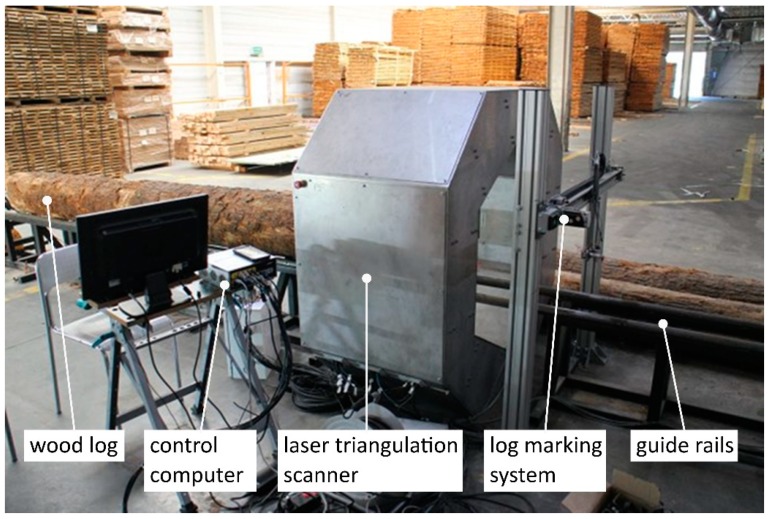
Log scanning system. In the picture the following elements of the system can be seen: scanned log placed on the conveyor, laser triangulation scanner, part of the log marking system, computer controlling the scanning process.

**Figure 8 sensors-19-01074-f008:**
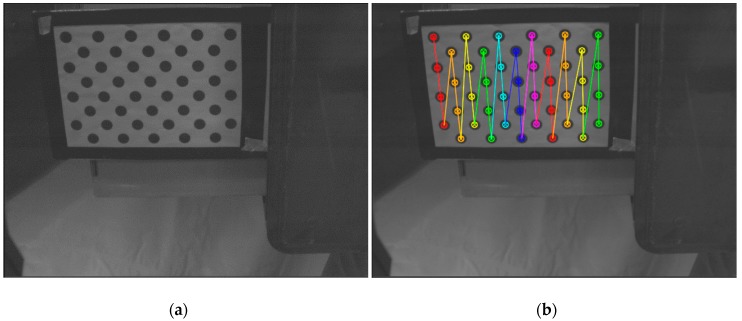
An exemplary image of the calibration pattern: (**a**) raw image; (**b**) image with extracted centers of the circles used to calibrate the camera.

**Figure 9 sensors-19-01074-f009:**
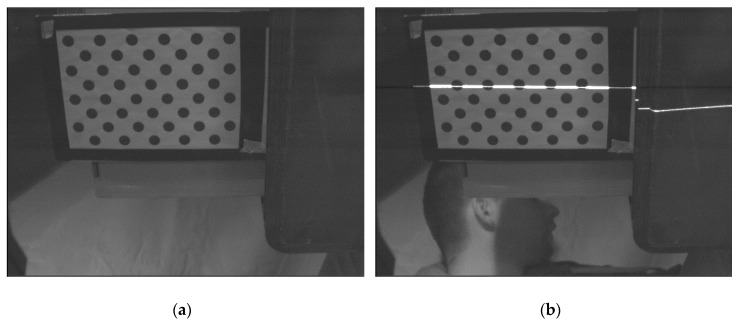
An exemplary image of a calibration pattern: (**a**) raw image; (**b**) image after switching on the laser.

**Figure 10 sensors-19-01074-f010:**
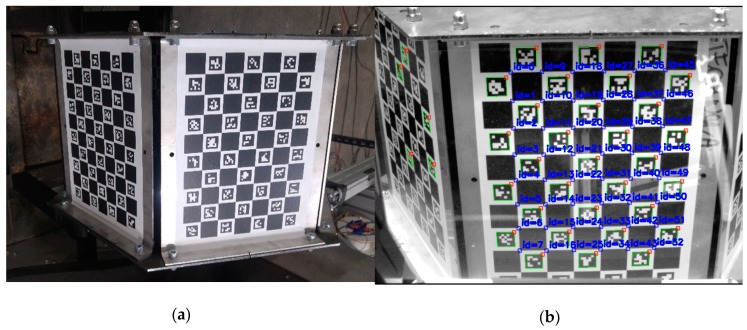
Hexagonal calibration artifact: (**a**) sample view; (**b**) detected markers on the artifact seen by the camera despite the reflections. Correctly detected markers are surrounded in green and calibration points (detected chessboard corners) are marked as blue squares.

**Figure 11 sensors-19-01074-f011:**
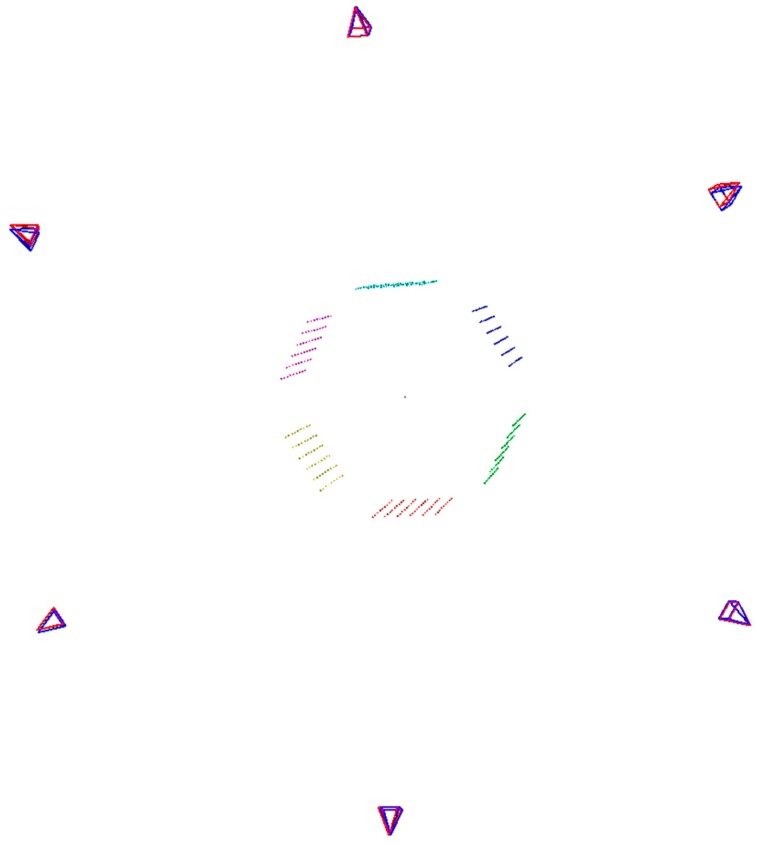
A symbolic representation of the calibration pattern and the determined positions of the cameras (red—determined directly through the image analysis; blue—after correcting the *x*-axis position). Each side of the calibration pattern is drawn in different color. The points represent the internal corners of each chessboard used in the global calibration procedure.

**Figure 12 sensors-19-01074-f012:**
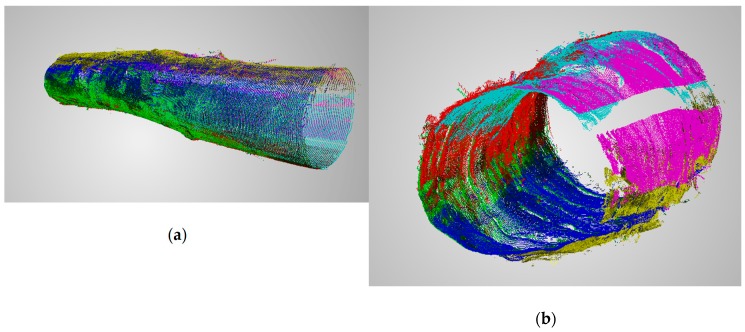
Exemplary effects of composing point clouds from different measuring heads. Data from different systems are marked with colors: (**a**) side view; (**b**) front view. Parts of magenta and yellow clouds are missing because they were occluded by the conveyor during measurement.

**Figure 13 sensors-19-01074-f013:**
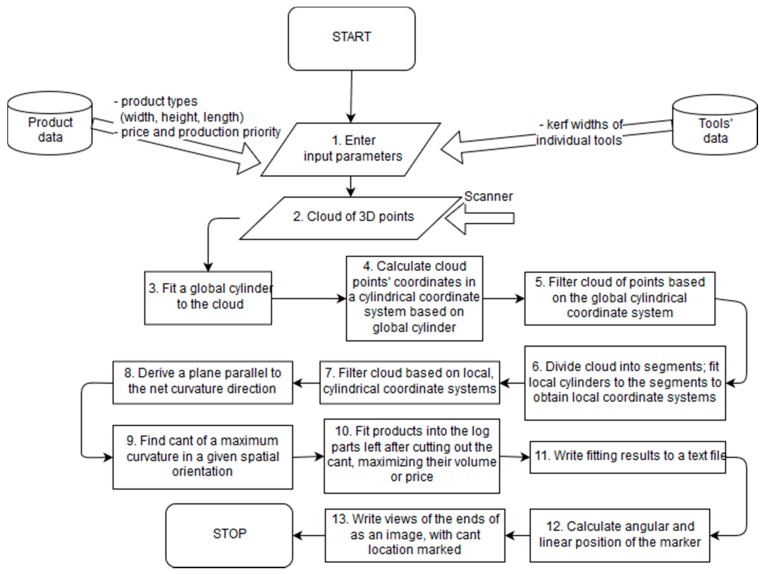
Flowchart of the data processing algorithm.

**Figure 14 sensors-19-01074-f014:**
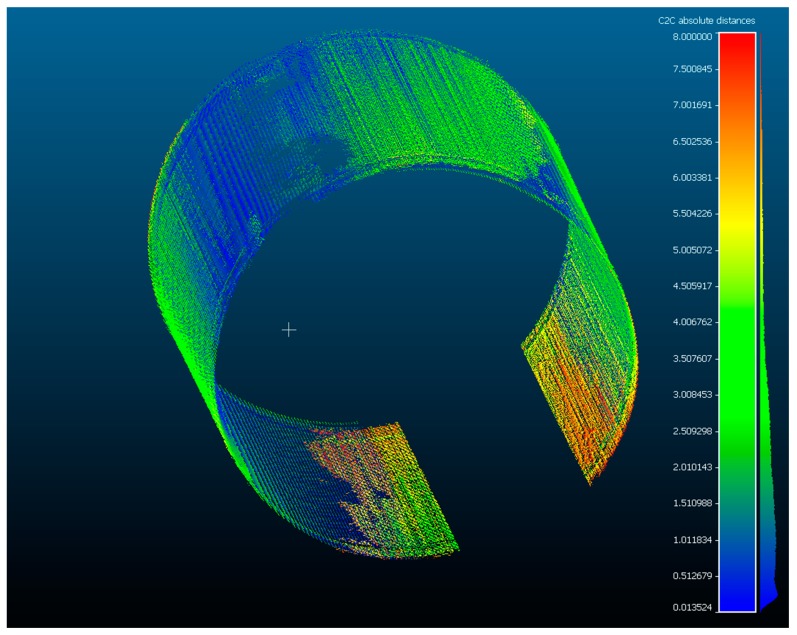
Visualization of the differences between the proposed scanner and the reference measurement on the example of a steel tube. The bottom of the tube was occluded during measurement.

**Figure 15 sensors-19-01074-f015:**
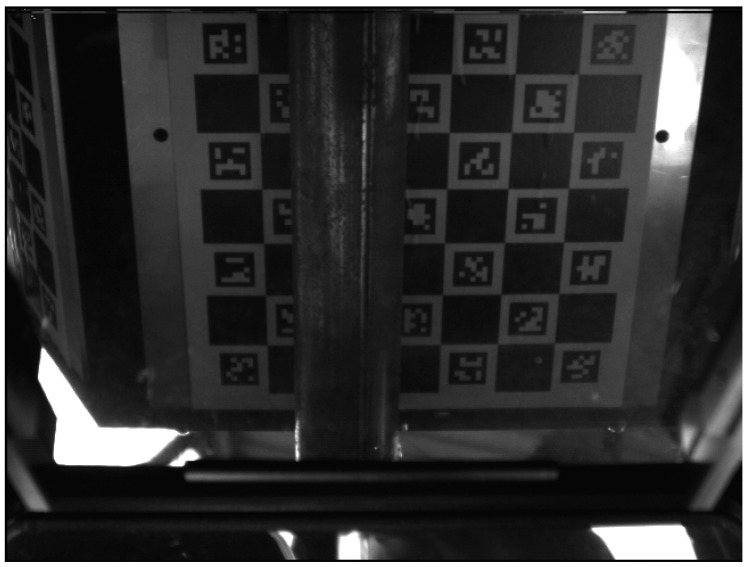
A view from the measurement head located at the bottom during the global calibration procedure. Some of the fields are occluded by the conveyor and therefore camera pose is estimated from a reduced set of chessboard corners and may be estimated sub-optimally.

**Figure 16 sensors-19-01074-f016:**
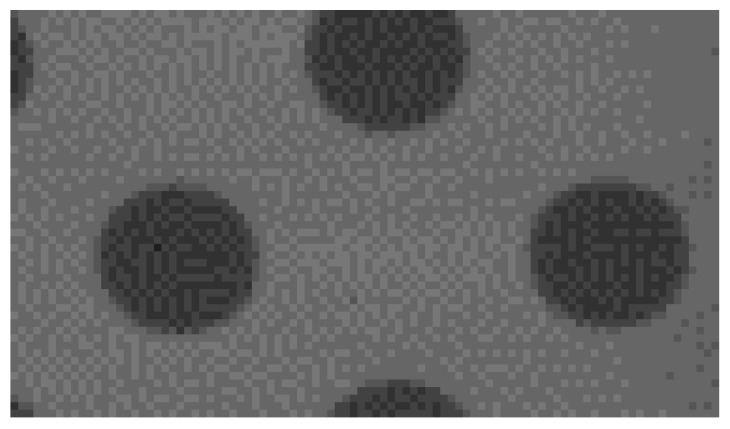
The close-up on the calibration pattern. A huge amount of noise is noticeable.

**Figure 17 sensors-19-01074-f017:**
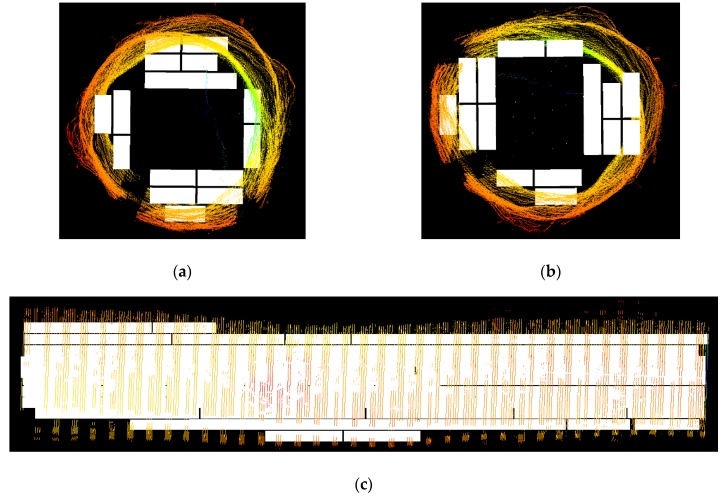
Sample log with the products fitted to the side parts of the log formed after cutting out the square cant. Simplified geometry of the log is drawn in colors ranging from green to red, depending on distance to the center axis of the log. Products are marked in white; (**a**,**b**): frontal views; (**c**,**d**): side views of the log.

**Table 1 sensors-19-01074-t001:** Comparison of the most important optical parameters of the methods, which can be used for scanning the shape of a surface.

Method	Structural Lighting	Laser Triangulation	ToF	Structure from Motion
**Surface**	Dispersive, partly non-transparent	Dispersive, non-transparent	Dispersive, non-transarent	Dispersive, non-transparent
**Resolution**	from 0.01 mm to 1 mm	0.01 mm	approx. 1 mm	0.1 mm
**Measuring range**	1 cm–2 m	2 mm–2 m	1 cm–a few km	A few cm–hundreds of km
**Accuracy**	1/1000 of a measuring field, max 0.005 mm	1/1000 of a measuring field, max 0.005 mm	Up to 1 mm	Depending on the examined object
**Color mapping**	Color detector required	Color detector required	Color detector required	Such as in 2D images
**Light conditions**	Additional lighting required	Additional lighting required	Additional lighting required	Natural light
**Measurement speed**	Average	High	Very high	Depends on the number of the images taken
**Expense**	Low	High	Average	Low
**Required equipment**	Projector, camera	Laser, camera	Laser, camera	Digital camera

**Table 2 sensors-19-01074-t002:** Local calibration accuracy evaluation of each system.

Measurement Head	Camera Reprojection Error [pixels]	Laser-Plane RMS [mm]
1	0.28	0.22
2	0.39	0.29
3	0.40	0.64
4	0.37	0.36
5	0.11	0.73
6	0.13	0.18

**Table 3 sensors-19-01074-t003:** The results of measurements of ball-bar in three different positions.

Ball-Bar Position	Parallel	Perpendicular	Oblique
**Standard deviation [mm]**	1.08	0.35	0.71
**Maximum error [mm]**	2.21	3.32	3.87

**Table 4 sensors-19-01074-t004:** The errors between measurements using the proposed scanner and the reference Metris MMCX80 laser scanner.

	Mean Error [mm]	Maximum Error [mm]	Min Error [mm]	Standard Deviation [mm]
Full cloud	2.41	8.02	0.01	1.83
Without measurements from the heads 3 and 4	1.86	5.13	0.01	1.49

**Table 5 sensors-19-01074-t005:** List of products fitted to the log.

Product Dimensions [mm]			Number of Pieces
Width	Thickness	Length	
170	31	1950	4
65	31	470	5
65	31	320	4
84	31	220	5
84	31	180	9
84	31	185	3
84	31	470	8
84	31	420	5
84	31	320	1
75	31	220	3
65	31	180	1
75	31	180	1
75	31	470	1
170	31	1930	1
65	31	420	3
65	31	185	2
65	31	220	2

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
