# Peer review of "On-Line Laser Triangulation Scanner for Wood Logs Surface Geometry Measurement"

_sensors, 2019, doi:10.3390/s19051074_

Round 1

Reviewer 1 Report

This paper demonstrated on-line laser triangulation scanner system consisting of 6 laser head to scan full surface shape of wood logs for sawmill track. The detailed comparisons and reviews for various methods of shape measurements would be good guideline to choose proper method for the use, which gives very useful insights for the readers. The detailed configuration of laser triangulation scanner and the investigation of the performance also be very interesting issues for the readers to measure some objects on conveyor line. Therefore, I recommend this paper can be published on Sensors as it is.

Author Response

Response to Reviewer 1 Comments

Authors of the manuscript would like to thank the reviewer for the review.

Point 1: This paper demonstrated on-line laser triangulation scanner system consisting of 6 laser head to scan full surface shape of wood logs for sawmill track. The detailed comparisons and reviews for various methods of shape measurements would be good guideline to choose proper method for the use, which gives very useful insights for the readers. The detailed configuration of laser triangulation scanner and the investigation of the performance also be very interesting issues for the readers to measure some objects on conveyor line. Therefore, I recommend this paper can be published on Sensors as it is.

Response 1: Thank you for this comment.

Reviewer 2 Report

1.Please explain the different types of errors in measurement system. 2. Please check the line 173(page 5)? 3. The description and mark of many figures ( Fig.2,4,6,7,8,10,11,12,16,17) are not clear enough. 4. Please explain these p1 and p2 of equation (3) and (4).

Author Response

Response to Reviewer 2 Comments

Authors of the manuscript would like to thank the reviewer for the review.

Point 1: 1.Please explain the different types of errors in measurement system.

Response 1: Thank you for this comment. We added another validation based on the custom-made ball-bar to determine measurement uncertainty.

Point 2: 2. Please check the line 173(page 5)?

Response 2: Thanks for spotting it. This should be a PI symbol. We corrected it.

Point 3: The description and mark of many figures ( Fig.2,4,6,7,8,10,11,12,16,17) are not clear enough.

Response 3: Thanks for this suggestion. We changed some of the figures to be more informative and we corrected multiple descriptions.

Point 4: Please explain these p1 and p2 of equation (3) and (4).

Response 4: We greatly appreciate this comment. p1 and p2 are tangential distortion coefficients We added this clarification to the manuscript.

Reviewer 3 Report

Interesting topic with industrial application. However, paper must be improved !!

Major comments:

1) literature review is focused only on measuring principles; there is no review of existing systems for measuring wood logs. Scientific papers, commercial systems, patents, should be revised and literature rewritten;

2) quality of pictures: there is no technical drawings tags, only screen shots, ... missing center lines, coordinate systems, units, annonations ...

3) The overall paper is written too general; scientific sound is missing;  math etc .. the only math (camera model) is generally known thing and could be cited, etc.

4) Measuring uncertainty? etc.

The text needs to be fully improved!

Author Response

Response to Reviewer 3 Comments

Authors of the manuscript would like to thank the reviewer for the review.

Point 1: Interesting topic with industrial application. However, paper must be improved !!

Response 1: Thank you for this comment. We added multiple clarifications based on the reviewers comments.

Point 2: literature review is focused only on measuring principles; there is no review of existing systems for measuring wood logs. Scientific papers, commercial systems, patents, should be revised and literature rewritten;

Response 2: Thank you very much for your comment. Indeed the literature overview in our paper is focused on the 3D surface shape measurement techniques that may be applied to investigate wood logs surface. Our intention was to show the deep insight and careful choice of the best possible solution for the needs of our business partner - Barlinek SA. We agree that there exist vast amount of scientific literature regarding wood measurement and processing, yet we could not find the specific paper describing 3D laser scanner, applied to wood log surface scanning, design and development, showing all components from the scratch. Therefore we decided to focus on this matter. Broad overview of existing systems for measuring wood logs including scientific papers, commercial systems and patents would make our paper more like review instead of focused scientific paper, which was our goal. Nevertheless we have slightly extended the introduction section to enrich the paper.

Point 3: quality of pictures: there is no technical drawings tags, only screen shots, ... missing center lines, coordinate systems, units, annonations ...

Response 3: Thank you very much for your comment. We added clarifications to multiple figures and their descriptions.

Point 4: The overall paper is written too general; scientific sound is missing;  math etc .. the only math (camera model) is generally known thing and could be cited, etc.

Response 4: We really appreciate this comment. The aim of the paper is to present this complex system, the method of its calibration and the methodology of the wooden log geometry measurements, which from our point of view is very interesting for applied scientists. We agree that the mathematic background (especially considering data processing procedure) is very important, it is a very extensive scientific material that we intend to describe in further article.

Point 5: Measuring uncertainty? etc.

Response 5: Thank you for this comment. We added another validation based on the custom-made ball-bar to determine measurement uncertainty.
